# Thermosensitive Chitosan Hydrogels: A Potential Strategy for Prolonged Iron Dextran Parenteral Supplementation

**DOI:** 10.3390/polym16010139

**Published:** 2023-12-31

**Authors:** Emerson Durán, Andrónico Neira-Carrillo, Felipe Oyarzun-Ampuero, Carolina Valenzuela

**Affiliations:** 1Departamento de Fomento de la Producción Animal, Facultad de Ciencias Veterinarias y Pecuarias, Universidad de Chile, Santa Rosa 11.735, La Pintana 8820808, Santiago, Chile; emerson.duran@ug.uchile.cl; 2Programa de Doctorado en Ciencias Silvoagropecuarias y Veterinarias, Campus Sur Universidad de Chile, Santa Rosa 11.315, La Pintana 8820808, Santiago, Chile; 3Laboratorios de Materiales Bio-Relacionados (CIMAT) y Síntesis y Caracterización de Polímeros Funcionalizados y Biomoléculas (POLYFORMS), Departamento de Ciencias Biológicas Animales, Facultad de Ciencias Veterinarias y Pecuarias, Universidad de Chile, Santa Rosa 11.735, La Pintana 8820808, Santiago, Chile; aneira@uchile.cl; 4Departamento de Ciencias y Tecnología Farmacéuticas, Facultad de Ciencias Químicas y Farmacéuticas, Universidad de Chile, Santos Dumont 964, Independencia 8380494, Santiago, Chile

**Keywords:** chitosan, thermosensitive hydrogel, iron deficiency, pig

## Abstract

Iron deficiency anemia (IDA) presents a global health challenge, impacting crucial development stages in humans and other mammals. Pigs, having physiological and metabolic similarities with humans, are a valuable model for studying and preventing anemia. Commonly, a commercial iron dextran formulation (CIDF) with iron dextran particles (IDPs) is intramuscularly administered for IDA prevention in pigs, yet its rapid metabolism limits preventive efficacy. This study aimed to develop and evaluate chitosan thermosensitive hydrogels (CTHs) as a novel parenteral iron supplementation strategy, promoting IDPs’ prolonged release and mitigating their rapid metabolism. These CTHs, loaded with IDPs (0.1, 0.2, and 0.4 g of theoretical iron/g of chitosan), were characterized for IM iron supplementation. Exhibiting thermosensitivity, these formulations facilitated IM injection at ~4 °C, and its significant increasing viscosity at 25–37 °C physically entrapped the IDPs within the chitosan’s hydrophobic gel without chemical bonding. In vitro studies showed CIDF released all the iron in 6 h, while CTH0.4 had a 40% release in 72 h, mainly through Fickian diffusion. The controlled release of CTHs was attributed to the physical entrapment of IDPs within the CTHs’ gel, which acts as a diffusion barrier. CTHs would be an effective hydrogel prototype for prolonged-release parenteral iron supplementation.

## 1. Introduction

Iron deficiency anemia (IDA) is the leading nutritional deficiency in the world, affecting one-third of the global population, especially during critical developmental stages such as childhood, pregnancy, and lactation [1,2]. This deficiency is caused by the high demand for iron during these stages and the low intake of iron in forms that provide higher bioavailability, such as those found in animal-sourced foods, which are expensive [3]. Multiple iron supplementation alternatives have been studied in humans and other mammalians to try and prevent anemia, with unsatisfactory results. In fact, iron deficiency has increased in the world. The use of low-bioavailability sources of non-heme iron (such as ferrous sulfate) in oral supplementation strategies and food fortification is the reason why these strategies have failed. Non-heme iron also causes adverse side effects like gastrointestinal disorders and has unpleasant sensory properties [2,4,5].

In humans, there are few options for parenteral iron supplementation formulations, mainly sodium ferric gluconate and intravenous (IV) iron sucrose. However, their use is not common due to the invasive nature of the procedure, risk of anaphylaxis, and high cost, as these procedures require hospitalization of the patient [6]. IV supplementation is highly effective for iron supplementation, achieving a more efficient increase in hemoglobin compared to oral supplementation, mainly due to its high bioavailability. However, there are risks of infections, anaphylactic reactions, endothelial damage, oxidative stress, and hepcidin overexpression [7].

In pigs, the IM supplementation of 150–200 mg of a commercial formulation of iron dextran (CIDF) in a single dose is used for the prevention of iron deficiency anemia [8], as the condition is highly prevalent in pigs raised in an intensive production system [9,10]. Pigs are considered a relevant and valid animal model to investigate iron deficiency and supplementation due to the physiological and metabolic similarities they have with humans and other omnivorous mammals [11,12]. CIDF is an aqueous dispersion of iron dextran particles (IDPs) with a ferric hydroxide core covered by a dextran shell, which imparts high hydrophilicity, low reactivity, and a nanometric particle size of ~11.5 nm [13]. When CIDF is injected into the muscular tissue, IDPs are rapidly dispersed across the surrounding tissues through simple diffusion mechanisms within muscle fibers, transported by the lymphatic system into the bloodstream, and captured by macrophages, which are responsible for extracting iron and binding it to transferrin for transport to the site of utilization or storage [8]. The preventive administration of CIDF to piglets has been employed for over 70 years. However, it exhibits several disadvantages linked to a rapid metabolism and a high amount of injected iron; a high iron load leads to a substantial increase in blood iron concentration within the first 10 h post-injection [14]. This triggers the overexpression of hepcidin, a hormone that promotes iron efflux from the body and reduces iron absorption, resulting in inefficient utilization of the supplemented iron [15,16]. Furthermore, the accumulation of such a substantial quantity of iron in storage sites induces toxic effects on adjacent tissues, due to its high reactivity potential [16,17].

To mitigate these adverse effects and enhance the effectiveness of injected IDPs, the use of chitosan thermosensitive hydrogels (CTHs) is proposed as a vehicle for the sustained release of IDPs. Chitosan is a biocompatible and biodegradable biopolymer that has been widely utilized to develop sustained release systems in various formats, including IM supplementation [18]. CTHs offer the advantage of being injected as a liquid (sol state) at room temperature (~4–25 °C) and significantly increase the viscosity, achieving a gel state, when interacting with muscular tissues (~36–37 °C) [19]. Several studies demonstrated the various benefits of CTHs for the controlled and/or prolonged release of active agents, which enhances drug retention in situ, thereby extending the duration of the release from the injection site [20,21]. In the present study, we postulate that CTHs possess the capacity to provide IDPs with in situ retention after IM injection. CTHs should act as a barrier, physically prolonging the release from the injection site and promoting a constant and low iron concentration in blood, and be able to prevent hepcidin overexpression and its negative consequences for iron supplementation.

Various molecules have been used to achieve thermosensitive behavior with chitosan, and this study focuses on the use of glycerophosphate (GP) due to its low toxicity, high accessibility, favorable thermosensitive behavior, and outcomes [18]. At room temperature (~4–25 °C), GP functions as an intermediary agent in the chitosan–GP–water system, enabling the maintenance of the sol state (with chitosan suspended) when the formulation is at a neutral pH. These interactions prevent chitosan from losing its cationic potential and settling due to its non-interaction with water [18]. However, as the temperature of the formulation raises to approximately 30 °C, the electrostatic chitosan–GP binding is disrupted, leading to chitosan–chitosan union through hydrogen bonding, which results in the formation of a highly viscous hydrophobic three-dimensional network (gel state) that impedes the outflow of the content from the injection site [18].

For these reasons, it is proposed to develop and study CTH-IDP formulations that maintain the sol state at room temperature (~4–25 °C) and undergo a transition to the gel state at a temperature close to the intramuscular (IM) temperature in mammals (~37 °C). The loading and entrapment of IDPs for prolonged iron supplementation through CTHs are investigated using a nanometric and hydrophilic active agent, and the outcomes of this study may serve as a model for the development of release studies involving IDPs or similar active agents. Therefore, the objectives of this study were to develop CTH-IDP formulations and to study their potential as a mammalian parenteral iron dextran supplementation strategy.

## 2. Materials and Methods

### 2.1. Materials

Iron dextran particles (IDP) obtained from a commercial formulation (CIDF, 20% *w*/*v* of iron, obtained from the same batch) were used as the iron source (Veterquímica S.A., Maipú, Chile). For the development of chitosan thermosensitive hydrogels (CTHs), chitosan derived from crab shell (300–350 KDa) with a degree of deacetylation >80% (Sigma-Aldrich, St. Louis, MO, USA) and hydrated disodium glycerophosphate (GP, Sigma-Aldrich, USA) were used. All other reagents were of analytical grade and procured from Merck S.A (Lethabong, South Africa).

### 2.2. Preparation of CTHs

The CTHs were developed following the procedure described by Sun et al. [22], with some modifications. A 0.2% *v*/*v* acetic acid solution in Milli-Q water at 4 °C was prepared, and 1% *w*/*v* chitosan was added, which was maintained at 4 °C for 12 h until a transparent and viscous solution (pH of 5.0–5.5) was obtained. GP was prepared at 50% *w*/*v* in Milli-Q water and added (0.2 mL/min) to the chitosan solution under magnetic stirring (4 °C), while the pH was constantly monitored until reaching neutral value (7.0 ± 0.5); the final GP concentration was ~6–8% *v*/*v*. The obtained CTHs were refrigerated until use.

After the CTHs were formed, IDPs in the form of CIDF were added (using a syringe at a rate of 1 mL/min) and homogenized using a paddle stirrer (1000 rpm for 1 h, OS40-PRO, D-LAB, Beijing, China). Three formulations with increasing iron concentrations (0.1, 0.2, and 0.4 g of theoretical iron/g of chitosan) were developed, referred to as CTH0.1, CTH0.2, and CTH0.4, respectively (CTH0, without iron, was the control).

### 2.3. CIDF Characterization

CIDF was characterized in terms of pH (AD1020, Adwa, Szeged, Hungary), particle size using dynamic light scattering (DLS, Zetasizer Nano-ZS90, Malvern Instruments, Malvern, UK), zeta potential by laser Doppler anemometry (Zetasizer Nano-ZS90, Malvern Instruments, UK), and viscosity using a rotational viscometer with a no. 1 needle (NDJ-8S, Nirun, Shanghai, China).

### 2.4. CTH Characterization

#### 2.4.1. Macroscopic Appearance

Digital images were obtained for macroscopic evaluation of the CTH0, CTH0.1, CTH0.2, and CTH0.4 formulations. The images were captured with the CTHs at room temperature (~22 °C) in test tubes from a focal point at 10 cm, focusing on the lower section of the tubes to visualize precipitates and aggregates.

#### 2.4.2. Electron Microscopy

Images of the CTH0, CTH0.1, CTH0.2, and CTH0.4 formulations in a gel state were obtained using scanning electron microscopy (SEM) to provide structural appreciation of the gel state. First, 1 mL of each gelled formulation (previously incubated at 37 °C for 30 min) was frozen at −80 °C and lyophilized (L101, Liotop, São Carlos, Brazil) for 24 h in Eppendorf tubes. Subsequently, the samples were coated with a 10 nm gold film using a Sputter Coater (Cressington model 108, Ted Pella Inc., Redding, CA, USA), and microscopic images were captured using a scanning electron microscope (FEI inspect F50, Thermo Fisher Scientific, Waltham, MA, USA) equipped with an energy dispersive detector (Ultradry Pathfinder Alpine 129 eV, Thermo Fisher Scientific, USA).

#### 2.4.3. pH

The pH of the CTH0, CTH0.1, CTH0.2, CTH0.4, and CIDF formulations was studied to assess compatibility with muscle tissue. The analysis was conducted on 100 mL samples using a standard pH meter (AD1020, Adwa, Szeged, Hungary).

#### 2.4.4. IDP Content

The concentration of IDPs in the CTH0.1, CTH0.2, CTH0.4, and CIDF formulations was measured using a UV spectrophotometer (UV-5100, Metash, Shanghai, China). A calibration curve for IDPs was obtained (λ = 486 nm, R^2^ = 0.99), providing a molar extinction coefficient of 2.9507 mL/mg·cm. This molar extinction coefficient was utilized for IDP quantification in CTHs. Based on this, the necessary dilutions were made for each formulation to achieve a theoretical concentration of 1 mg/mL of IDPs. Results were presented as mg of IDP/mL of formulation.

#### 2.4.5. Viscosity and Sol–Gel Transition Time

To confirm and analyze the thermosensitivity of the CTH0, CTH0.1, CTH0.2, and CTH0.4 formulations, the viscosity was determined in 200 mL samples at 4, 25, and 37 °C using a rotational viscometer (NDJ-8S, Nirun, China). All measurements were carried out with a no. 1 needle, and the unit of measurement used was milliPascal-second (mPa·s). The formulations were incubated (BJPX-200B, Biobase, Jinan, China) for 30 min prior to each experiment, which was conducted at room temperature (~20 °C) immediately following incubation.

To determine the time necessary for the sol–gel transition for these formulations at 37 °C, the tube inversion method described by Wang et al. [23] was used. First, 5 mL of sample was transferred to a sealed glass test tube (13 mL capacity, 1.5 mm diameter) and kept at 4 °C. Simultaneously, a beaker containing distilled water was incubated at 37 °C. The experiment consisted of submerging three-quarters of the test tube into the beaker and measuring the time it takes for the formulation to reach the gel state. The sample was considered to be in a gel state when, upon rotating the tube 180°, the formulation did not flow. The sol–gel transition was monitored every 30 s.

#### 2.4.6. Water–Gel Phase Separation

In order to determine if IDPs are entrapped within them, CTHs, CTH0.1, CTH0.2, and CTH0.4 formulations were centrifuged in the gel state to induce separation between the hydrophobic gel and the aqueous phase of the formulation, thereby revealing the position of the IDPs through macroscopic digital images. Additionally, the same experiment was carried out on the CTH0 and CIDF formulations: 30 mL of the formulations was incubated in Falcon tubes at 37 °C for 30 min and then centrifuged at 3000× *g* for 30 min using a Thermo Scientific Heraeus Megafuge 16R centrifuge (USA). After centrifugation, the macroscopic appearance of the formulations was assessed in the same manner described in the section on macroscopic appearance.

#### 2.4.7. Fourier-Transform Infrared Spectroscopy

An analysis of the infrared spectrogram was performed on CTH0, CTH0.1, CTH0.2, and CTH0.4 formulations, as well as the chitosan polymer and CIDF (dried at 50 °C for 48 h in an oven), to understand the predominant bonds in the formulations/precursor materials and how IDP content influences the bonds. Fourier-transform infrared spectroscopy (FTIR) analysis was conducted using an ATR/FTIR instrument (Interspec 200-X spectrometer, Tartumaa, Estonia). The formulations were incubated at 37 °C for 30 min before measuring to ensure they were in a gel state. Spectra were obtained by averaging 20 scans in the spectral range of 600–4000 cm⁻¹.

#### 2.4.8. Injectability and Retention Evaluation in Ex Vivo Porcine Tissue

The selected formulations (CTH0.4, due to its higher IDP content, and CIDF as a control) were ex vivo injected to confirm they can be injected in porcine tissue in sol state using a syringe. Digital images were obtained to evaluate and compare the retention of each formulation in the injection site. The selected porcine tissue was top inside round, corresponding to the semitendinosus muscle of the hind limb (common CIDF IM injection site in pigs), of an approximate size of 5 (width) × 5 (length) × 3 (height) cm^3^. The porcine tissue was purchased from a supermarket, then sized, and kept refrigerated to be used within the next 12 h. Tissue samples were injected to a depth of 1 cm (using a 3 mL plastic syringe attached to a G21 needle, with an internal diameter of 0.60 mm) with 1 mL of each formulation. The injected tissue samples were then incubated at 37 °C for approximately 1 h, to simulate the temperature of a mammalian in vivo muscle tissue. The images were captured from a focal point at 10 cm, after injection and incubation. To expose the injection site and visualizing the retention of the formulation, the tissue was transversely cut.

#### 2.4.9. In Vitro Iron Release

The quantification of the In vitro iron release from the selected CTH formulation (CTH0.4) and CIDF, a release study using a USP 4 apparatus (Sotax CE7 smart, CY 7 piston pumps, Sotax, Westborough, MA, USA) was carried out. Experiments were conducted in triplicate, utilizing phosphate buffered saline (PBS) as the release medium within each cell (100 mL). A consistent flow rate of 8 mL/min was maintained, and the temperature was regulated at 37 °C. Each cell was equipped with a sample holder containing 200 µL of the formulation, directly positioned above the beads. To minimize turbulence, a 6 mm diameter glass bead was placed at the bottom of each cell, surmounted by a 2 cm layer of 1 mm of diameter glass beads. Sampling was performed at designated intervals (0.5, 1, 2, 4, 6, 8, 12, 24, 48, and 72 h), withdrawing 1 mL of release medium, which was subsequently replaced with the same volume of fresh PBS. Iron content in CTH0.4, CIDF, and collected samples was determined using a Perkin Elmer PINAACLE 900 F Atomic Absorption Spectrophotometer, equipped with an acetylene/air flame for iron quantification. Sample preparation involved digestion with nitric acid and hydrogen peroxide, heating at 90 °C for 2 h. The results were represented as mean values on a cumulative content release curve over time. In vitro drug release data were exposed to mathematical kinetics models (program DDSolver). The Akaike information criteria (AIC), coefficient of determination (R^2^), and the model selection criteria (MSC) values were considered for the selection of the model, and then the release data were fitted to different models (zero order, first order, Higuchi, and Korsmeyer–Peppas).

#### 2.4.10. Statistical Analysis

Characterizations of pH, sol–gel transition time, viscosity, IDP content, particle size, zeta potential, and In vitro iron release generated results with continuous and normal data (Shapiro–Wilk test, *p* > 0.05). Characterizations were carried out in triplicate. To determine significant differences, ANOVA (*p* < 0.05) and Tukey’s test (*p* < 0.05) were used. Calculations were performed using R software version 4.3.1 (R package, Boston, MA, USA). FTIR analysis was conducted through graphical representation for better visualization and comparison. Macroscopic appearance, electron microscopy, water–gel separation, and injectability analyses were completed using the obtained images.

## 3. Results and Discussion

### 3.1. CIDF Characterization

CIDF was characterized by pH and viscosity, which are properties of interest for understanding the use of CIDF as an IM supplement. The pH was 6.38 ± 0.03, which is considered suitable for IM use in mammals (including pigs and humans), as they show a physiological pH close to 7. Therefore, CIDF is unlikely to cause pain upon injection due to the activation of pH-sensitive receptors [24,25]. The viscosity of CIDF significantly decreased with temperature: 23.8 ± 0.2 mPa·s at 4 °C, 12.9 ± 0.2 mPa·s at 25 °C, and 10.2 ± 0.5 mPa·s at 37 °C. This change can be explained by the progressive breaking of hydrogen bonds between the IDPs and the water in the formulation as the energy in the system increases due to the effect of temperature. Therefore, when injected, the temperature of the muscle decreases the viscosity of the CIDF, facilitating its dispersion in the adjacent tissues. The obtained viscosity is in line with the values obtained by other authors (10–25 mPa·s) and is considered suitable for extrusion using a needle with a 21–25 G lumen, commonly used in IM injection in pigs, humans, and other mammals [26].

The average particle size of IDPs in CIDF was 81.9 ± 0.2 nm, which differs from what was described [27], where a particle size of 11.5 nm was obtained using the same method employed in the present study (DLS). This size for these IDP particles prevents diffusion directly into the bloodstream and metabolization through the lymphatic system, acting as a barrier that provides a delayed iron absorption [8]. However, IDP use delays iron absorption but does not add a prolonged or sustained release, resulting in elevated serum iron concentrations within the first 1–10 h post-injection [14]. The zeta potential of IDPs was −0.15 ± 0.56 mV, representing a neutral surface charge value, implying low potential for interaction with membranes and molecules bearing more significant electrical charges [28]. A neutral charge increases the potential for particle aggregation, as they do not electrostatically repel each other [29], which is advantageous for retention within a CTH, promoting accumulation and stability at the injection site.

### 3.2. CTH Characterization

#### 3.2.1. Macroscopic Appearance

All developed formulations are shown in Figure 1a. Those containing IDPs exhibit a brown/orange coloration, which is attributed to the presence of iron hydroxide in the IDP cores. At room temperature (~22 °C), it is evident that all formulations show a heterogeneous appearance. Moreover, the formation of small clusters can be observed by the naked eye, which may result from the premature gelation of CTHs mediated by nucleation–aggregation processes, where chitosan–chitosan bonds, established via hydrogen bridges at multiple points that simultaneously grow, form small hydrophobic aggregates that increase the formulation’s viscosity [18]. The presence of aggregates is initially considered a disadvantage for extrusion through a syringe with a 21–25 G needle due to potential obstructions. Additionally, if this gelation process is triggered at lower temperatures than 20 °C, it also represents a potential disadvantage due to the non-uniform distribution of the active ingredient, which could impede dose uniformity; therefore, the formulations should be injected at lower temperatures.

#### 3.2.2. Electron Microscopy

SEM images of the lyophilized formulations are presented in Figure 1b. The surfaces of the formulations containing IDPs appear to be more heterogeneous than those without it (CTH0), which is most noticeable in the formulation with a higher IDP content (CTH0.4), which also has a rougher surface. This could be attributed to the presence of IDPs interrupting the spatial distribution of CTHs, leading to more discontinuous materials. In an interesting study, Zhao et al. [30] obtained images of CTHs in the gel state (without active principles added) prepared with different types of acid to protonate the amino groups of chitosan, including the same acid used in the present study (acetic acid). They observed highly heterogeneous structures, which align with the simultaneous aggregation of chitosan–chitosan in multiple cores as the temperature increases in CTHs [18]. EDS analysis revealed that the lyophilized CTH0, CTH0.1, CTH0.2, and CTH0.4 formulations exhibit a high presence of oxygen, phosphorus, carbon, and sodium, which are characteristic elements of chitosan and GP. With this strategy, the presence of iron is also observed in the CTH0.1, CTH0.2, and CTH0.4 formulations (~10–20% *w*/*w*), indicating the retention of this element in lyophilized CTHs. This could occur through the physical entrapment of IDPs within the CTHs, allowing retention even when water is extracted. Therefore, the three-dimensional networks formed in the gel state of CTHs are heterogeneous due to the gelation of CTHs mediated by the nucleation–aggregation process and the presence of IDPs, being capable of retaining the iron content after water extraction.

#### 3.2.3. pH

The pH is an important property for the biocompatibility of CTHs, since a change in the proton concentration at the injection site can cause pain in the animal due to the activation of pH-sensitive receptors [24,25]. As shown in Table 1, all CTH formulations have similar pH values and are close to neutrality. These results demonstrate that the addition of IDPs does not have an acidifying effect on the formulations, obtaining pH values close to the physiological range for pigs, humans, and other mammals (~7–7.4) [31]. Therefore, the obtained CTHs show a pH suitable for use as an IM supplement for pigs and other mammals, not requiring pH rectification.

#### 3.2.4. IDP Content

The IDP content of the CTH formulations is presented in Table 1 and is a direct consequence of the initially added CIDF content (0.1, 0.2, and 0.4 g of theoretical iron/g of chitosan). The IDP content in CTH0.4 is significantly higher than in the other CTH formulations. The iron content in the formulations (in the form of IDPs) could be increased to approach the CIDF content; however, this would result in an excessive increase in the viscosity, due to the high content of IDPs, and a decrease in the concentration of thermosensitive molecules in the formulation.

#### 3.2.5. Viscosity and Sol–Gel Transition Time

The thermosensitivity of CTHs is the most relevant characteristic for IM use, as it enables the smooth injection of the formulation in a sol state and subsequent transition to the gel state within the muscular tissue. As shown in Table 1, the obtained viscosity values demonstrate thermosensitivity in all developed formulations, with similar values at 4 and 25 °C across all groups. However, as the temperature increases to 37 °C, all CTHs exhibit significantly higher and similar viscosity values. Importantly, the transition to the gel state occurs independently of the added IDP content.

The thermosensitivity of these formulations is explained by a series of chemical reactions triggered by the increase in the formulation temperature. At neutral pH, it is not possible to solubilize chitosan in water because it is necessary to protonate its amino groups for the molecule to acquire a polar character and interact with water. However, the addition of GP allows chitosan to not precipitate at neutral pH and remain in a sol state at room temperature (~4–25 °C). This is because GP molecules act as intermediaries in the chitosan–GP–water interaction at neutral pH, allowing chitosan to remain suspended as long as the GP–chitosan interaction between the phosphate and amino groups is maintained. This interaction also prevents the deprotonation of chitosan at neutral pH because its amino groups are shielded by the hydration shell formed by the GP–water interaction [18]. However, increased temperature (higher than 25 °C) promotes the definitive transfer of protons from amino groups to phosphate ions, losing the chitosan–GP interaction, leading to the formation of chitosan–chitosan interactions through hydrogen bonds due to the reduction in chitosan interchain electrostatic repulsion [32]. This proton transference is a direct consequence of a thermosensitive drop in the pKa of chitosan (~−0.025 pK units/°C), promoting neutralization and losing its cationic potential [33]. This chitosan–chitosan union generates non-reversible three-dimensional networks with a nonpolar character, which macroscopically translates into an increase in viscosity, known as the gel state [18].

The significant increase in viscosity between 25 and 37 °C, as observed in Table 1, is advantageous because it allows the formulations to be injected in the sol state close to room temperature (~<22 °C), which then substantially increases the viscosity within the muscle tissue. If increased fluidity in IM injection is required, the CTH temperature could be reduced close to 4 °C. This aligns with that previously described in the macroscopic appearance section, where, at room temperature (~22 °C), the formulations appeared viscous and exhibited the formation of small aggregates. This could indicate that the gelation process begins at temperatures <22 °C, with a peak occurring at temperatures higher than 25 °C. The maximum values obtained by the formulations in this study are similar to those reported by [30], ranging from ~2000 to 5000 MPa·s in CTHs synthesized with different materials.

The determination of the time for CTHs to reach the gel state at muscular temperature (~37 °C) is crucial for the development of a potential prolonged-release IDP supplement because the release of the content is likely to be higher prior to the gel state, due to the absence of a force opposing the diffusion of the compound at the injection site. As observed in Table 1, the sol–gel transition time of the CTH formulations increased with the IDP content, being three times longer in CTH0.4 compared to the control. This phenomenon suggests that the reactions responsible for increasing the viscosity require more time to occur in the presence of IDPs and aligns with the fact that the final viscosity values were similar for all formulations (Table 1). This could mean that the formation of a three-dimensional chitosan–chitosan network responsible for the gel state occurs independently of the IDP content, delaying the thermosensitivity but not limiting it. This delay in the transition time could be explained by the presence of non-thermosensitive IDPs and their temporal interference in the chitosan–chitosan binding that leads to the gel state. Finally, these results are consistent with findings from other authors, where transition times of approximately 1–10 min were observed; in those works, the molecular weight and concentration of chitosan were higher [19]. In conclusion, the proposed formulations are thermosensitive, the sol–gel transition time is sensitive to the IDP content, and the tested CTHs show values within appropriate limits for potential IM injection.

#### 3.2.6. Water–Gel Phase Separation

The effective entrapment of IDPs within CTHs in the gel state is necessary to achieve sustained release. To demonstrate this entrapment, the formulations were centrifuged in the gel state to separate the hydrophobic gel and the water in the formulation, forcing the release of IDPs with water since they are highly hydrophilic. This was completed to determine whether the entrapment of IDPs within the chitosan network is sufficient to prevent escape. In Figure 2, the formulations in the gel state before (Figure 2a) and after (Figure 2b) centrifugation are shown. All formulations were separated into two phases, except for CIDF, which keeps the IDPs suspended in water because of its high hydrophilicity. In formulations with phase separation, the upper phase corresponds to the hydrophilic portion of the hydrogel (white arrow, Figure 2b), mainly consisting of GP suspended in water. The lower phase (black arrow, Figure 2b) corresponds to the hydrophobic gel, which was completely separated from the water during centrifugation due to higher density. It is noted that in CTHs containing IDPs, the lower phase (gel) retains the coloration of the initial formulation, indicating that the IDPs (previously suspended) remain in the gel after centrifugation, unlike CIDF. Considering that a chemically attractive interaction between IDPs and chitosan is unlikely due to the respective hydrophilic/hydrophobic characteristics; it is hypothesized that this phenomenon is likely the result of the physical entrapment of IDPs in the chitosan gel. In summary, the action of the CTHs seems to allow IDP retention in chitosan hydrophobic gel at 37 °C.

#### 3.2.7. Fourier-Transform Infrared Spectroscopy

The identification of IDP–chitosan interactions and the possible formation of new chemical bonds can help determine the nature of IDP entrapment and anticipate the characteristics of release at the injection site. In Figure 3, the spectra of CTHs in the gel state and the materials used for synthesis (CIDF and chitosan as dry powders) are presented. First, in CTH formulations, an absorption band is observed in the region between 3700 and 3200 cm^−1^, primarily reflecting O-H bonds from the water molecules highly present in these formulations and N-H interactions from the amino functional group of chitosan [34]. CIDF and CTH formulations show two absorption bands around 3000 cm^−1^, corresponding to C-H bonds in the aliphatic CH_2_ and CH_3_ groups present in the structures of chitosan and dextran [34]. CTH formulations exhibit an absorption band near 1650 cm^−1^, corresponding to the vibrations of C=O bonds in the NH_2_ amino group of chitosan monomeric units and O-H bonds from water molecules [34]. In the 1200–1550 cm^−1^ range, CTH formulations, CIDF, and chitosan display absorption bands, mostly corresponding to the characteristic vibrations of O-H, C-H, and C-O bonds inherent to chitosan and dextran structures [34,35]. Likewise, in CTH formulations, within the range of 800–1200 cm^−1^, there are absorption bands corresponding to -O- and P-O-C bonds characteristic of chitosan and GP structures, respectively [34]. These data appear to reveal that the bonds present in CTH/IDP formulations exhibit absorption bands characteristic of CTHs without IDPs, and there is no evidence of the formation of new bonds or the disappearance of previously existing bonds and specific signals of iron–chitosan interactions, as described by Fahmy and Sarhan et al. [36]. In CT/IDP formulations, there was no observed reduction in the intensity of the absorption band in the 3700–3200 cm^−1^ region or the absorption band near 1650 cm^−1^, indicating that the bonds of chitosan amino groups show no differences with the addition of iron, and there was no evident appearance of/variation in the characteristic absorption bands of Fe-N or Fe-O interactions [36]. The formation of IDP–chitosan interactions through dextran is unlikely due to dextran’s low reactivity, attributed to steric hindrance from its functional groups, and the loss of the cationic potential of chitosan in the gel state. However, it is not possible to rule out the formation of hydrogen bonds between IDPs and chitosan [29]. CTHs containing IDPs exhibit bonds characteristic of CTHs, and there is no evidence of chitosan–IDP interaction.

#### 3.2.8. Injectability and Retention Evaluation in Ex Vivo Porcine Tissue

The injectability of the formulations was confirmed through an ex vivo injection test. It was possible to inject the formulation with the higher iron content (CTH0.4) and the gold standard treatment (CIDF) into porcine tissue incubated at 37 °C (Figure 2c,d). CIDF was poorly retained in the injection site (Figure 2c), while CTH0.4 (Figure 2d) was efficiently retained. The low retention of CIDF in injection site can be explained by its high hydrophilicity and low viscosity, which facilitate the rapid diffusion through muscle fibers from the injection site. In contrast, CTH0.4 retention is probably a consequence of the CTH sol–gel transition after its intimate contact with the muscle tissue preheated to 37 °C, triggering the gel formation. The high viscosity in the gel state at 37 °C (Table 1) should promote the retention of IDPs in the injection site. Furthermore, contrasting these results with the water–gel phase separation, it is hypothesized that after the injection of CTHs, the chitosan polymer network forms a viscous gel, concentrating the IDP content at the injection site and prolonging the iron release.

#### 3.2.9. In Vitro Iron Release

With the aim of studying the IDP release from CTHs, we selected a validated methodology commonly used to study formulations with prolonged release (USP apparatus 4). This methodology consists of the supply of a continuous flux of a selected medium in a cell containing the formulation. Although this methodology mimics the dynamic of fluids better than conventional dialysis for parenteral formulations, it is also more demanding due to the constant flux of the medium exposed to the formulation. The release of IDPs from CIDF and CTH0.4 was investigated by quantifying the elemental iron released at 37 °C. Iron release from CIDF reached 100% within 6 h, in contrast to the CTH0.4 formulation, which achieved 40% release over 72 h (Figure 4). This prolonged release from CTH0.4 is attributed to the sol–gel transition of the chitosan thermosensitive hydrogel at the medium’s temperature (37 °C), forming a hydrophobic gel that reduces IDP diffusion. In contrast, the rapid release from CIDF is due to the nanometric size and hydrophilic nature of IDPs, facilitating their quick diffusion. CTH gelation creates an effective polymeric barrier that confines IDPs, leading to their prolonged release. The brief duration of iron release observed in this study is consistent with the reported maximum iron concentrations in porcine blood within the first 12 h following CIDF administration [14], which can be attributed to the quick release and subsequent metabolism of IDPs.

The release mechanisms of the CIDF and CTH0.4 formulations were elucidated by fitting the release data to various models (Table 2). The Korsmeyer–Peppas model emerged as the most suitable, indicated by the highest correlation coefficient (R^2^), the lowest Akaike information criterion (AIC), and the highest model selection criterion (MSC) [37]. The diffusional exponent (n) values for both CIDF and CTH0.4 (both below 0.5) suggest Fickian diffusion as the predominant release mechanism. This aligns with the rapid liberation observed for CIDF in Figure 4, attributed to constant diffusion driven by the IDP concentration gradient [38]. Conversely, the initial rapid release, followed by a slower release pattern of IDPs from CTH0.4, as shown in Figure 4, implies a similar initial release mechanism to CIDF for non-entrapped IDPs, with a posterior slower release of entrapped IDPs from the gel matrix. The behavior of CTH0.4 could be considered as positive, the initial quick diffusion release may be used for the immediate iron requirements, and the prolonged iron release from the gel could be useful to maintain the biological effect over the time.

Figure 5 proposes the interaction dynamics between the different components of the formulation (chitosan, GP, IDPs, and water) in sol and gel states. In the sol state (Figure 5a), the dominant forces are the interactions of water–GP and water–IDP hydrogen bonds, which keep these components suspended, in addition to the electrostatic GP–chitosan bond that prevents the precipitation of the polymer as it is anchored to water through the GP [18]. When transitioning to the gel state (Figure 5b), the increase in temperature generates a decrease in the cationic potential of chitosan, resulting in the loss of the chitosan–GP interaction and the generation of chitosan–chitosan hydrogen bonds, which give origin to the high-viscosity hydrophobic network [18]. Based on the results obtained by and discussed in the present study, it is proposed that IDPs are confined in the chitosan network, which is a physical barrier that stands in the way of diffusion, prolonging IDPs’ liberation. Finally, based on all the characterizations included in this article, future studies need to be focused on improving and optimizing the potential shortcoming of our strategy and carrying out an in vivo study in pigs. These improvements should focus on increasing the iron load, improving the CTH homogeneity, and increasing the gelation temperature to avoid any transition at <25 °C, to ensure an adequate injectability in the sol state. In contrast, the irreversible thermosensibility for the obtained materials makes it difficult to use them in high-temperature regions (>25 °C) because the materials can be transformed to the gel state before the injection; in those cases, the CTHs can be refrigerated and injected at a lower temperature.

## 4. Conclusions

CTHs loaded with IDPs were successfully synthesized, obtaining thermosensitive formulations with neutral pH, suitable for IM injection in a sol state at approximately 4 °C and transitioning into a gel state between 25 and 37 °C. It was determined that IDPs are effectively entrapped within the chitosan polymer network in the gel state, without chemical interactions between chitosan–IDP or GP–IDP. In vitro release studies revealed that CIDF released 100% of its iron content within 6 h, while CTH0.4 released 40% over 72 h, predominantly through Fickian diffusion. The pronounced difference in release profiles was primarily due to the physical confinement of IDPs within the CTH gel, which acted as an additional diffusion barrier. These findings contributed valuable insights on CTHs as a strategy for prolonged iron supplementation and laid the groundwork for developing new prolonged-release micronutrient supplementation approaches for pigs, humans, and other mammals.

## Figures and Tables

**Figure 1 polymers-16-00139-f001:**
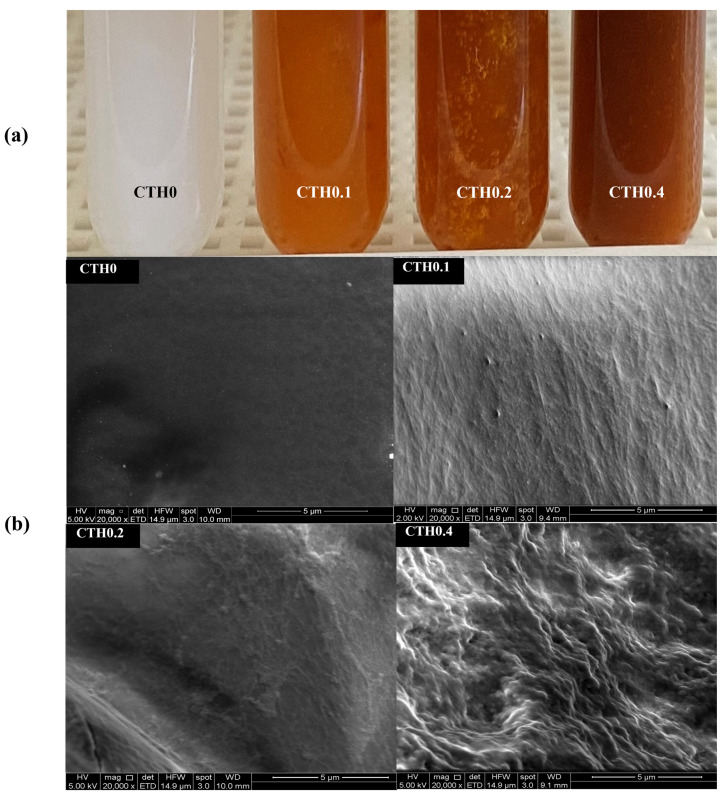
Macroscopic appearance (**a**) and electron microscopy images (**b**) of the thermosensitive chitosan/iron dextran hydrogel formulations (CTHs) with increasing iron concentrations (0.1, 0.2, and 0.4 g of theoretical iron/g of chitosan).

**Figure 2 polymers-16-00139-f002:**
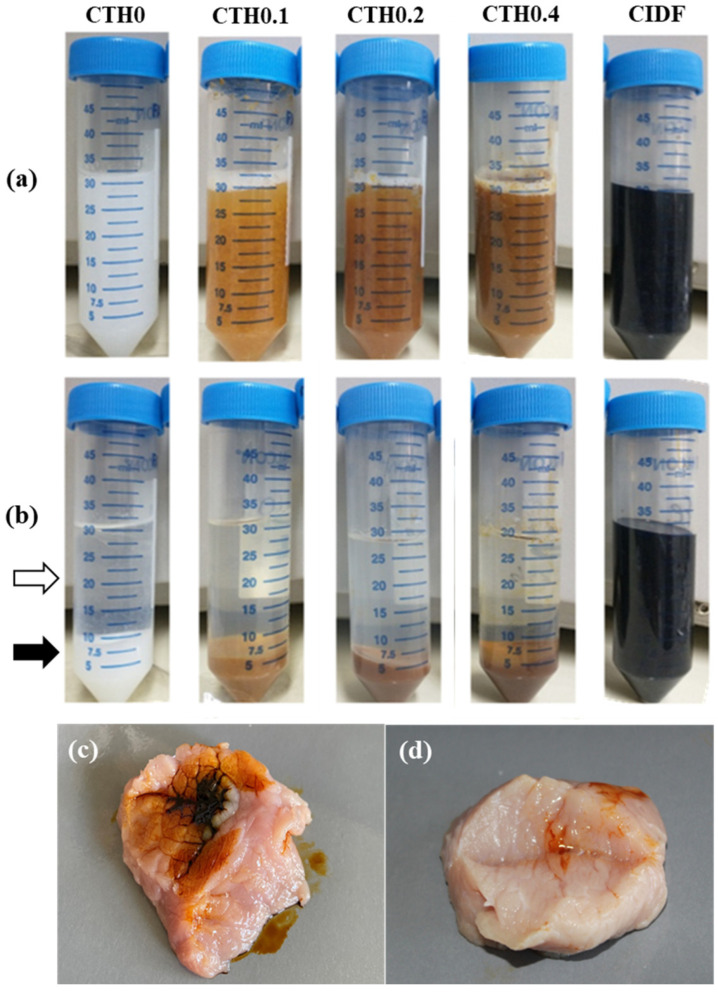
Macroscopic appearance of the chitosan thermosensitive hydrogel (CTH) formulations with increasing iron concentrations (0.1, 0.2, and 0.4 g of theoretical iron/g of chitosan) and the commercial iron dextran formulation (CIDF), showing the separation of the sol–gel phases at 37 °C before (**a**) and after (**b**) centrifugation. The white and black arrows indicate the upper (water) phase and the lower (gel) phase of CTH0, respectively; (**c**,**d**) show the appearance of CIDF and CTH0.4 injected at 37 °C into a piece of pork meat, respectively.

**Figure 3 polymers-16-00139-f003:**
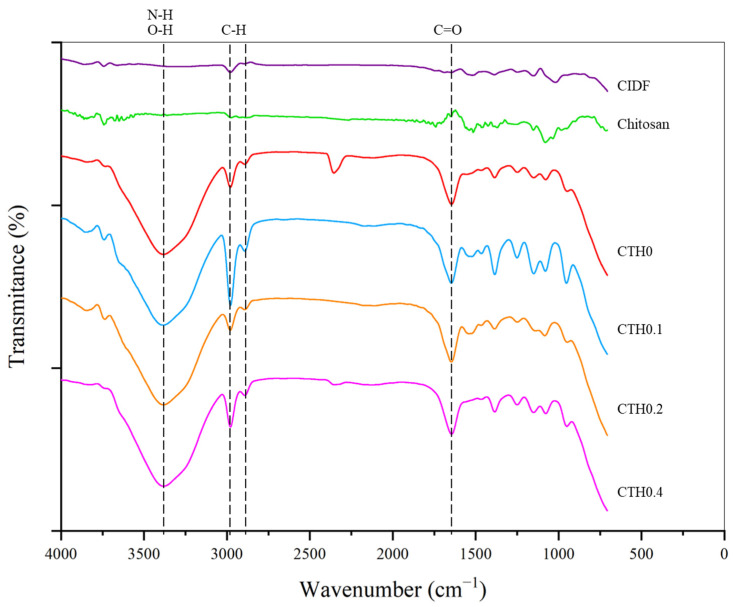
Spectrograms of the chitosan thermosensitive hydrogel (CTH) formulations with increasing iron concentrations (0.1, 0.2, and 0.4 g of theoretical iron/g of chitosan), the commercial iron dextran formulation (CIDF), and precursors. Dotted lines mark the main absorption bands. Data obtained through infrared spectroscopy.

**Figure 4 polymers-16-00139-f004:**
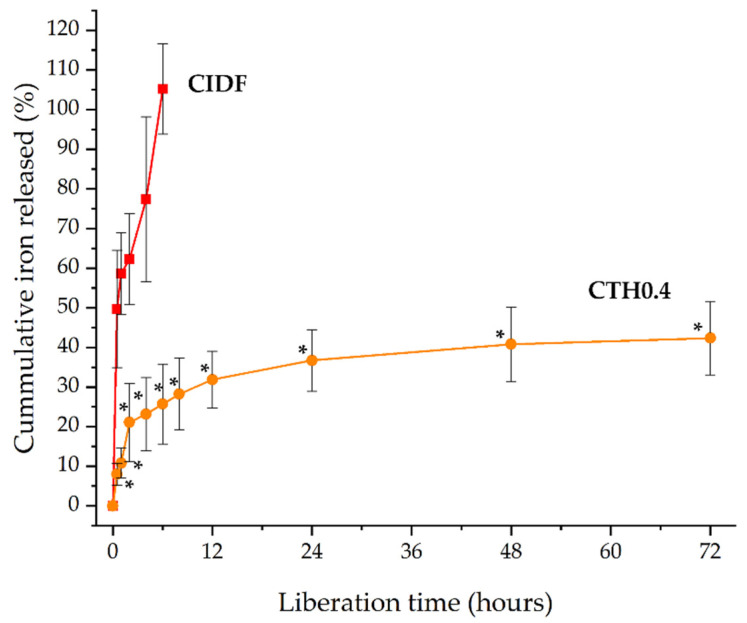
Cumulative iron release from commercial iron dextran formulation (CIDF) and the chitosan thermosensitive hydrogel containing iron dextran particles (CTH0.4) at 37 °C. * Significative differences between formulation values for each measurement time (*p* < 0.05).

**Figure 5 polymers-16-00139-f005:**
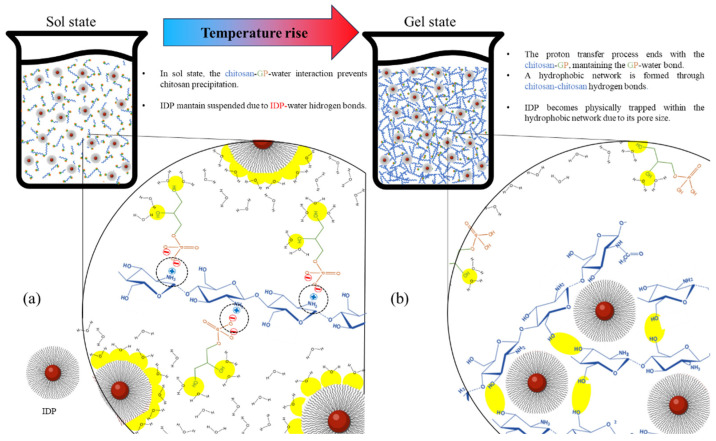
Two-dimensional schematic representation of interactions among chitosan, glycerophosphate (GP), water, and the iron dextran particles (IDPs) in a chitosan thermosensitive hydrogel in sol state (**a**) and gel state (**b**). Dotted circles represent chitosan–GP ionic bonds, and yellow areas represent interactions involving GP–water, chitosan–chitosan, and water–IDP hydrogen bonding.

**Table 1 polymers-16-00139-t001:** pH, sol–gel transition time, iron dextran particle (IDP) content, and viscosity at 4, 25, and 37 °C of the chitosan thermosensitive hydrogel (CTH) formulations with increasing iron concentrations (0.1, 0.2, and 0.4 g of theoretical iron/g of chitosan) and CIDF.

Formulation	pH	Sol–Gel Transition Time (s)	IDP Content(mg/mL)	Viscosity (MPa·s)		
				4 °C	25 °C	37 °C
CTH0	6.88 ± 0.07	90 ^a^	-	45 ± 10 ^a^	65 ± 13 ^a^	2925 ± 108 ^b^
CTH0.1	6.83 ± 0.07	120 ^b^	2.0 ± 0.3	72 ± 15 ^b^	383 ± 33 ^b^	2807 ± 284 ^b^
CTH0.2	6.62 ± 0.01	120 ^b^	4.0 ± 0.8	74 ± 9 ^b^	269 ± 40 ^c^	3052 ± 421 ^b^
CTH0.4	6.68 ± 0.07	300 ^c^	13 ± 2	134 ± 14 ^c^	447 ± 13 ^b^	3060 ± 151 ^b^
CIDF	6.38 ± 0.03	-	-	23.8 ± 0.2 ^a^	12.9 ± 0.2 ^a^	10.2 ± 0.5 ^a^

Different letters (^a–c^) indicate significant differences between CTH and CIDF formulations for each of the characterizations (*p* < 0.05).

**Table 2 polymers-16-00139-t002:** Kinetic model parameters for CIDF and CTH0.4 release data.

Formulation	Release Model	R^2^	AIC	MSC	n
CIDF	Zero order	0.30	52.00	−0.70	
	First order	0.80	44.73	0.50	
	Higuchi	0.81	44.60	0.53	
	Korsmeyer–Peppas	0.88	40.94	1.14	0.27
CTH0.4	Zero order	−0.48	88.92	−0.92	
	First order	−0.17	86.40	−0.69	
	Higuchi	0.60	73.93	0.44	
	Korsmeyer–Peppas	0.91	60.88	1.62	0.33

## Data Availability

Data are contained within the article.

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
