# Peer review of "Thermosensitive Chitosan Hydrogels: A Potential Strategy for Prolonged Iron Dextran Parenteral Supplementation"

_polymers, 2023, doi:10.3390/polym16010139_

Round 1

Reviewer 1 Report

Comments and Suggestions for Authors

1.    The hypothesis of this manuscript is to develop a vehicle for sustained release of IDP, but the authors didn’t include any data about the release study. Authors need to do some sustained release study.

2.    As the author mentioned, CTH were loaded with IDP at increasing concentration, what' the loading efficacy, what about the saturation?

3.    In table 1, authors mentioned different letters indicate significant differences in each characterization (P<0.05), authors need to go more details about the definition of “a, b, c”.

4.    Authors mentioned that the IDP are not sufficient due to the quick metabolization, so, CTH was developed. Does the CTH address the metabolization issue? Authors need to investigate that. Moreover, authors did so many of characterization, it would be better to have some in vitro or in vivo study to move it forward as potential strategy for parenteral nutrient supplementation.

5.    The title of this manuscript is really confused, which need to improve.

6.    Why was GP used for preparing the thermosensitive behavior? Did authors compare with other molecular to confirme?

7.    Figure 2 c and d, authors did it in pork meat, instead of using the live pig tissue. It would be better if authors include some live animal study.

8.    IDP was physically trapped in the chitosan hydrophobic gel without evidence of chemical bonding, what’s the potential mechanism related to this?

9.    What about the potential shortcoming of this hydrogel? Authors need to discuss more related to the shortage of this strategy.

Comments on the Quality of English Language

Moderate editing of English language required

Author Response

Reply to reviewers.

Dear Editor, please find below our reply to reviewers. All necessary changes, in order to address concerns of the reviewers, have been included in the revised version and highlighted in red. We believe that the suggested comments have significantly improved the quality of the manuscript.

Reviewers' comments

Reviewer #1

1) Reviewer comment: The hypothesis of this manuscript is to develop a vehicle for sustained release of IDP, but the authors didn’t include any data about the release study. Authors need to do some sustained release study.

Answer: We agree with the reviewer, so we include an in vitro release experiment using the USP apparatus 4 (Figure 4). USP apparatus 4 is one of the most recognized and validated methods to study in vitro prolonged.

2) Reviewer comment: In table 1, authors mentioned different letters indicate significant differences in each characterization (P<0.05), authors need to go more details about the definition of “a, b, c”.

Answer: As the reviewer required, details of the assigned letters indicating significant differences were provided at the bottom of Table 1. The statistics were also reanalyzed, as there were some errors.

4) Reviewer comment: Authors mentioned that the IDP are not sufficient due to the quick metabolization, so, CTH was developed. Does the CTH address the metabolization issue? Authors need to investigate that.

Answer: This was investigated through an in vitro release study, a hypothesis related with this aspect is also proposed. This is included in the revised version of this manuscript (Figure 4).

5) Reviewer comment: Moreover, authors did so many of characterization, it would be better to have some in vitro or in vivo study to move it forward as potential strategy for parenteral nutrient supplementation.

Answer: As mentioned above, an in vitro release study was added (Figure 4).

6) Reviewer comment: The title of this manuscript is really confused, which need to improve.

Answer: The title was improved to be more understandable.

7) Reviewer comment: Why was GP used for preparing the thermosensitive behavior? Did authors compare with other molecular to confirm?

Answer: This explanation was added.

8) Reviewer comment: Figure 2 c and d, authors did it in pork meat, instead of using the live pig tissue. It would be better if authors include some live animal study.

Answer: This manuscript is a preliminary study covering the development and characterization of these potential iron vehicles for parenteral application, we enriched the original manuscript with an in vitro release study. Therefore, in vivo studies are the subject of a forthcoming publication.

9) Reviewer comment: IDP was physically trapped in the chitosan hydrophobic gel without evidence of chemical bonding, what’s the potential mechanism related to this?

Answer: To answer this question, we analyzed and compare the in vitro release data with zero-order, first-order, Korsmeyer-Peppas, and Higuchi liberation models; the Korsmeyer-Peppas model fits better. We conclude that, within this model, our data suggest a Fickian diffusion mechanism for both formulations, at least at 72 h. These new data were added to the manuscript.

10) Reviewer comment: What about the potential shortcoming of this hydrogel? Authors need to discuss more related to the shortage of this strategy.

Answer: The shortcomings of these hydrogels were addressed in the corrected version of this manuscript.

Reviewer 2 Report

Comments and Suggestions for Authors

The authors have made an interesting article describing the thermosensitive chitosan hydrogels as a vehicle for iron dextran as a potential strategy for parenteral nutrient supplementation but some improvements should be made:

  1. Line 155 it should be explained and detailed “the yielding of the specific molar extinction coefficient in water of 2.9507 mL/mg·cm (slope 155 of the curve)”
  2.  Line 194 and Line 425 please rephrase “Injectability in pork meat”
  3. Line 200 “cm3” written with superscript.
  4. Line 426 ex vivo with Italic
  5. The injectability in pork meat should be quantified in a scientifically matter not only put the images after the injection took place. Please elaborate this test with quantified parameters that should be interpreted.

Author Response

Reviewer #2

 1) Reviewer comment: Line 155 it should be explained and detailed “the yielding of the specific molar extinction coefficient in water of 2.9507 mL/mg·cm (slope 155 of the curve)”.

Answer: We added more details in the manuscript.

2) Reviewer comment: Line 194 and Line 425 please rephrase “Injectability in pork meat”.

Answer: We agree with the reviewer and changed it for “Injectability and retention evaluation in ex vivo porcine tissue”.

3) Reviewer comment: Line 200 “cm3” written with superscript.

Answer: The change was done.

4) Reviewer comment: Line 426 ex vivo with Italic

Answer: The change was done.

5) Reviewer comment: The injectability in pork meat should be quantified in a scientifically matter not only put the images after the injection took place. Please elaborate this test with quantified parameters that should be interpreted.

Answer: The objective of the porcine tissue characterization was just to show that the formulation can be injected with a syringe and its distribution from the injection site in a simple and intuitive way. Quantification of the iron release differences between commercial iron dextran formulation (CIDF) and the chitosan thermosensitive hydrogel containing iron dextran particles (CTH0.4) was addressed in an in vitro release study included in the corrected version of this article (Figure 4).

Please, do not hesitate in contact us if any additional clarification is required.

Kind regards,

Prof. Carolina Valenzuela V.

[email protected]

Universidad de Chile.

Felipe Oyarzun-Ampuero

[email protected]  

Universidad de Chile

Round 2

Reviewer 1 Report

Comments and Suggestions for Authors

Accept in current version.